# DisA Limits RecG Activities at Stalled or Reversed Replication Forks

**DOI:** 10.3390/cells10061357

**Published:** 2021-05-31

**Authors:** Rubén Torres, Carolina Gándara, Begoña Carrasco, Ignacio Baquedano, Silvia Ayora, Juan C. Alonso

**Affiliations:** Department of Microbial Biotechnology, Centro Nacional de Biotecnología, CNB-CSIC, 28049 Madrid, Spain; rtorres@cnb.csic.es (R.T.); carolina.gandara@newcellsbiotech.co.uk (C.G.); bcarrasc@cnb.csic.es (B.C.); ignacio.baquedano@ua.es (I.B.); sayora@cnb.csic.es (S.A.)

**Keywords:** DNA repair, stalled fork, RecG, DisA, c-di-AMP, template switching

## Abstract

The DNA damage checkpoint protein DisA and the branch migration translocase RecG are implicated in the preservation of genome integrity in reviving haploid *Bacillus subtilis* spores. DisA synthesizes the essential cyclic 3′, 5′-diadenosine monophosphate (c-di-AMP) second messenger and such synthesis is suppressed upon replication perturbation. *In vitro*, c-di-AMP synthesis is suppressed when DisA binds DNA structures that mimic stalled or reversed forks (gapped forks or Holliday junctions [HJ]). RecG, which does not form a stable complex with DisA, unwinds branched intermediates, and in the presence of a limiting ATP concentration and HJ DNA, it blocks DisA-mediated c-di-AMP synthesis. DisA pre-bound to a stalled or reversed fork limits RecG-mediated ATP hydrolysis and DNA unwinding, but not if RecG is pre-bound to stalled or reversed forks. We propose that RecG-mediated fork remodeling is a genuine in vivo activity, and that DisA, as a molecular switch, limits RecG-mediated fork reversal and fork restoration. DisA and RecG might provide more time to process perturbed forks, avoiding genome breakage.

## 1. Introduction

Replicative DNA polymerase(s) may stall at obstacles (DNA lesions, RNA polymerases trafficking conflicts, etc.), and then recombination functions are required to support replication restart [1,2,3]. Cells have evolved multiple mechanisms to prevent incorrect handling of perturbed replication forks that *per se* do not compromise genome integrity. For instance, a replisome can simply skip a lesion in the lagging-strand template, leaving single-stranded (ss) DNA regions coated by the single-stranded binding protein (bacterial SSB/SsbA or eukaryotic RPA). An unrepaired lesion on the leading-strand template transiently uncouples the replicative DNA helicase from the stalled replicative DNA polymerase(s) and causes ssDNA regions coated by SSB/SsbA/RPA. This enables to circumvent the lesion by several DNA damage tolerance (DDT) sub-pathways [1,2,3,4,5].

When DNA replication was hindered in *Escherichia coli* by mutations in some replication proteins, it was observed that the stalled replication fork reversed, with subsequent annealing of the nascent strands [5,6,7]. Replication fork reversal (also named fork regression), i.e., the active conversion of a stalled replication fork into a Holliday junction (HJ) structure, has emerged as a global and genetically controlled response to aid in the repair or bypass of DNA damage during replication stress [5,8]. In *E. coli*, the enzymes that process stalled forks are RecA, RuvAB, RecG and RecQ [9,10,11,12]. Here, the regressed arm of the reversed fork, which resembles a one-ended double-strand break (DSB), can be degraded by nucleases as the RecBCD complex (counterpart of *Bacillus subtilis* AddAB), or the HJ-like structure can be processed by the RuvABC complex, leading to fork breakage and repair by homologous recombination [13,14]. Since during the first replication cycle of a reviving haploid *B. subtilis* spore end resection functions have not been synthesized yet [15,16] and there is only one copy of the replicating genome, a reversed fork (mimicking a one-ended DSB) may represent a pathological structure that can be lethal. Therefore, such fork remodeling should be regulated.

A powerful tool to characterize the functions involved in replication fork remodeling and how they are controlled is to damage the DNA of inert non-replicating haploid *B. subtilis* spores with ionizing radiation and revive spores synchronously under unperturbed conditions. In the absence of both end resection pathways (i.e., in the ∆*recJ* ∆*addAB* strain), those reviving spores remain recombination proficient and are apparently as capable of repairing DNA damages as the wild-type (wt) control [17], suggesting that reviving spores activate other repair mechanisms in response to ionizing radiation treatment prior revival. Indeed, the ionizing radiation-induced broken ends can be reconnected by the Ku and LigD proteins via non-homologous end-joining [18], and the nicks can be repaired via a LigD-dependent pathway [19] during the ripening stage of spore revival. Later, at the early stage of outgrowth, the expression of *ku* and *ligD* is turned off, and that of DNA replication and repair proteins involved in single-strand gap repair and DDT is turned on [15,16]. Here, replication fork impediments (damaged template bases, DNA distortions, replication-transcription conflicts) can be circumvented or bypassed by the action of RecA accessory proteins (e.g., RecO, RecR, RecF, RecX), RecA itself, the branch migration translocase RecG and the DNA damage checkpoint protein DisA, among others [17,20]. The rationale is that, as a global response to replication stress in the absence of long-range end resection functions and an intact homologous template [15,16,17], as it occurs during the first replication cycle of reviving spores, it is necessary to use a different DDT mechanism (fork reversal, template switching, translesion synthesis) to avoid uncontrolled spore revival, that otherwise could be deleterious. (Unless stated otherwise, indicated genes and products are of *B. subtilis* origin).

The diadenylate cyclase (DAC) DisA converts a pair of ATP molecules into cyclic 3′, 5′-diadenosine monophosphate (c-di-AMP) [21], an essential second messenger that regulates a wide variety of physiological functions, [22], including the monitorization of replication perturbations. Single-cell analyses revealed that DisA, acting as a DNA damage checkpoint, shows a dynamic movement along the chromosome during sporulation, but it pauses upon DNA damage until the offending lesion is removed [23]. DisA pauses at a cognate site formed at or downstream RecA engagement to DNA lesions that compromise replisome progress [24]. Paused DisA drops the synthesis of c-di-AMP to levels comparable to that in the absence of DisA in response to replication perturbations, as those induced by methyl methane sulfonate (MMS) [25,26]. Indeed, DisA binding to DNA substrates that mimic branched intermediates suppresses c-di-AMP synthesis, but not when bound to double-stranded (ds) DNA [21,27], placing the primary function of DisA at a stalled or reversed fork to circumvent or bypass offending lesions. In this context, it has also been observed that: (i) in unperturbed exponentially growing cells, DisA shows a dynamic movement in wt cells, but it becomes static in the absence of the RecG branch migration translocase (where branched recombination intermediates accumulate) [27]; and (ii) the *disA* and *recG* genes show an epistatic interaction upon DNA damage [27].

The RecG translocase is ubiquitous among Bacteria, land plants, green algae and moss (targeted to mitochondria and chloroplast) [28,29]. RecG homologues have not been detected in archaea and other eukaryotes. Here, the mammalian SMARCAL1 translocase may be its functional ortholog [30]. RecG is a multi-functional protein and there is conflicting evidence on its mode of action [29]. In vitro, *E. coli* RecG (RecG*_Eco_*) promotes branch migration of a stalled replication fork to form a HJ-like structure, and it can also branch migrate the resulting HJ back to a fork structure (fork restoration), as the SMARCAL1 enzyme does [30,31,32,33,34]. In the presence of a reconstituted replication system, it is observed that the reversed forks formed by RecG*_Eco_*, rather than by RuvAB*_Eco_*, are the substrate for RuvC*_Eco_* (counterpart of *B. subtilis* RecU) cleavage [35]. Other studies proposed a division of labor with RecG*_Eco_* acting at DSBs, and the RuvAB*_Eco_* translocase acting at stalled forks reviewed in [7].

The role of RecG in other bacteria remains poorly understood because not all the activities associated to RecG*_Eco_* are present in other distant RecG proteins [29]. For instance, in vivo, inactivation of *recG* shows disparate phenotypes in the different bacteria tested: (i) a *Helicobacter pylori* Δ*ruvAB* Δ*recG* mutant strain shows no significant difference with the wt control in response to DNA damage [36]; (ii) an *E. coli* Δ*ruvAB* Δ*recG* mutant strain has a severe DNA repair defect [37]; and (iii) a Δ*recG* mutation is synthetically lethal in the Δ*ruvAB* context in *B. subtilis* and *Neisseria gonorrhoeae* [38,39]. Finally, there are biochemical activities associated with RecG*_Eco_* that are not observed in *B. subtilis*: (i) RecG*_Eco_* binds and unwinds a variety of branched DNA substrates, including HJs, D-loops and R-loops, but unwinding of R-loops was not seen with *B. subtilis* RecG [40]; and (ii) RadD*_Eco_* bound to abandoned fork structures modulates RecG*_Eco_* activities [41], but a RadD-like enzyme is absent in *B. subtilis* cells. These observations suggest that both RecG enzymes may work at forked DNA structures, but the information obtained with RecG*_Eco_* may not be easily extrapolated to other bacteria, highlighting the vital importance of revision in science.

A possible role for RecG at stalled and reversed forks in concert with DisA was earlier proposed based on genetic evidence [20,27], and here we explore their biochemical interplay. We show that RecG preferentially branch migrates stalled forks with damage in the lagging-strand and reversed forks, but it also reverses stalled forks with damage in the leading-strand, though with slightly lower efficiency, as documented for RecG*_Eco_*. DisA does not form a stable complex with RecG, but they mutually affect their activities depending on the order of protein addition. DisA pre-bound to stalled or reversed forks downregulates RecG activities. DisA-mediated c-di-AMP synthesis is additively inhibited by RecG and HJ DNA. We propose that DisA may downregulate RecG-mediated processing of branched intermediates to prevent unscheduled and toxic DNA intermediates, that could lead to fork breakage. DisA might limit RecG activities to favor lesion bypass or template switching to quickly convert the gaps in duplex DNA. This would allow the removal of offending lesions by the specialized repair machineries on haploid genomes, followed by replication re-initiation.

## 2. Materials and Methods

### 2.1. Protein-Protein Interaction Assays

In vivo protein-protein interaction was assayed using the adenylate cyclase-based bacterial two-hybrid assay, as earlier described [42]. The *E. coli* BTH101 cells bearing plasmids pUT18, pUT18C, pKNT25 and pKT25, and their derivatives, were used. The plasmid-borne DisA fusions to the T18 or T25 catalytic domain of the adenylate cyclase, either at the N- (DisA-T18 and DisA-T25) or C-terminus (T18-DisA or T25-DisA) were previously described [42]. These plasmids were pairwise co-transformed into the reporter BTH101 strain with plasmid-borne RecG fusions, also to the T18 or T25 catalytic domain, either at the N- or the C-terminus. The empty plasmids or the pKT25-zip and pUT18C-zip vectors were co-transformed into the reporter strain as negative and positive controls, respectively. Different dilutions were spotted onto LB plates supplemented with ampicillin, kanamycin, streptomycin, 0.5 mM IPTG (Calbiochem, Madrid, Spain), and 10% X-Gal (Merck KGaA, Darmstadt, Germany). The plates were then incubated at 25 °C for 3–4 days.

In vitro protein-protein interactions were assayed using affinity chromatography as earlier described [43]. His-tagged DisA (1 µg), RecG (0.5 µg) or both proteins were pre-incubated in buffer B (50 mM Tris-HCl pH 7.5, 1 mM DTT, 50 mM NaCl, 1 mM MgCl_2_, 0.05 mg/mL BSA, 5% glycerol) (5 min, 37 °C) and then loaded onto a 50 µL Ni^2+^ matrix microcolumn, and the flow-through (FT) was collected. After extensive washing, the retained proteins were eluted with 50 µL of Buffer B containing 400 mM imidazole. The proteins were separated by 10% SDS-PAGE, and gels were stained with Coomassie Blue (Merck KGaA, Darmstadt, Germany).

### 2.2. DNA Substrates and Protein Purification

All chemicals used were analytical grade. DNA restriction enzymes and DNA ligase were from New England Biolabs (Ipswich, MA, USA).

The nucleotide (nt) sequences of the oligonucleotides used are indicated in Appendix A. A fixed HJ (HJ-J3) was constructed by annealing J3-1, J3-2, J3-3 and J3-4; the mobile HJ (HJ-J4) by annealing J170, J173, J345 and J346; the 3′-tail HJ DNA by annealing J3-1, J3-2-110, J3-3 and J3-4; the 5’-tail HJ DNA by annealing J3-1, J3-2-110-5, J3-3 and J3-4; the Y-fork by annealing J170, 171, 172 and J173; the 3´-fork by annealing J170, 172 and J173; the 5´-fork by annealing J170, 171 and J173; a flayed by annealing J170 and J173; the forked-Lag by annealing J170, J173, 171 and 172-15; and the forked-Lead by annealing J170, J173, 171-15 and 172. The ssDNA concentrations were measured using the extinction coefficient of 1.54 × 10^-4^ M^−1^ cm^−1^ at 260 nm, and the concentrations of ssDNA or HJ DNA are expressed as moles of DNA molecules or moles of nucleotides as indicated. The radiolabeled annealed products were gel purified as described previously, dialyzed against buffer A (50 mM Tris-HCl (pH 7.0), 1 mM MgCl_2_, 5% glycerol, 1 mM DTT) and stored at 4 °C [44].

*E. coli* BL21(DE3) cells bearing pLysS and pET-derived plasmids were used for protein overexpression and XL1-Blue cells were used for plasmid amplification. *E. coli* plasmids pCB951, pCB875, pCB1080, pCB1081 and pCB1229 were used to over-express the RecG, DisA, DisA D77N DisA ∆C290, and PcrA proteins, respectively [24,27,45].

DisA, DisA D77N, DisA ∆C290 and RecG were purified as previously described [24,27,45]. Purified PcrA was provided by M. Moreno-del Álamo (CNB-CSIC). The purified proteins were >97% pure based on staining after SDS-PAGE, and partial proteolysis and MALDI-TOF analysis. RecG was free of RecG*_Eco_* protein. The molar extinction coefficients for DisA, RecG and PcrA were calculated as 22,350, 63,260 and 70,375 M^−1^cm^−1^ at 280 nm as described [46]. The concentrations of RecG, DisA and PcrA are expressed as moles of monomers, octamers and monomers, respectively.

### 2.3. DNA Binding

DNA binding was assayed by electrophoretic mobility shift assays (EMSAs). The binding of the indicated protein, alone or in the presence of a second protein, to the synthetic [γ^32^P]-HJ-J3 DNA (0.1 nM in molecules) was performed in buffer B containing 5 mM ATPγS. Reactions were incubated for 15 min at 37 °C. Prior to loading, 0.2% glutaraldehyde was added to stabilize the complexes. Protein-DNA complexes were separated using 6% PAGE in TAE buffer and visualized by autoradiography. Autoradiography films were scanned, and the ImageJ software (NIH, Maryland, MD, USA) was used to determine the signal from each band and obtain the apparent binding constant (K_Dapp_) values.

The reaction conditions for DNase I footprint experiments were as for EMSA. DNase I treatment was performed in Buffer C (50 mM Tris-HCl pH 7.5, 1 mM DTT, 50 mM NaCl, 10 mM MgCl_2_, 0.05 mg/mL BSA, 5% glycerol) as described [47]. The samples were resuspended in loading buffer [80% (*v*/*v*) formamide, 10 mM NaOH, 1 mM EDTA, 0.1% (*v*/*v*) bromophenol blue, and 0.1% (*v*/*v*) xylene cyanol], separated in 15% denaturing PAGE (dPAGE) and autoradiographed.

### 2.4. Helicase Assays

In a standard branch migration assay [γ^32^P]-labelled DNA (0.4 nM) was incubated with the first protein (RecG or DisA) in buffer C for 5 min at 37 °C. Then, the second protein (DisA, RecG or no protein) and 5 mM ATP were added, and the reaction was incubated for 15 min more. Reactions were terminated by adding one-fifth volume of stop mix (5% SDS, 100 mM EDTA, 5 mg/mL proteinase K) and further incubation for 10 min at 37 °C to deproteinize the sample. Unwound products were analyzed by 6% PAGE in TAE buffer. Radioactive gels were exposed to storage phosphor screens and scanned using a PMI Molecular Imager (Bio-Rad, California, USA). Signals of substrates and products were quantified with the Image Lab (Bio-Rad, California, USA) software. The statistical significance of the data was analyzed by applying t-tests.

### 2.5. ATPase Activity and c-di-AMP Synthesis

The DNA-dependent ATP hydrolysis activity of the RecG protein was assayed by monitoring the disappearance of absorbance at 340 nm, due to NADH conversion to NAD, using a Shimadzu CPS-20A (Tokyo, Japan) dual-beam spectrophotometer via an ATP/NADH-coupled spectrophotometric enzymatic assay. The DNA effectors were a 3199-nucleotides (nt) circular pGEM3 Zf(+) ssDNA (cssDNA) or HJ-J3 DNA (10 µM in nt). Assays were done in buffer D (50 mM Tris-HCl pH 7.5, 80 mM NaCl, 10 mM MgOAc, 0.05 mg/mL BSA BSA, 1 mM DTT, 5% glycerol) containing 5 mM ATP and an ATP regeneration system (620 μM NADH, 100 U/mL of lactate dehydrogenase, 500 U/mL pyruvate kinase, and 2.5 mM phosphoenolpyruvate) for 30 min at 37 °C. The concentration and order of addition of the purified proteins are indicated in the text. Data obtained from A_340_ absorbance were converted to ADP produced, and plotted as a function of time. In some reactions, the curves reached a plateau, due to a consumption of the NADH pool, and the rate of ATP hydrolysis was derived from the slope of the linear part of the curves as described [48]. T-tests were applied to analyze the statistical significance of the data.

c-di-AMP synthesis by DisA was analyzed using thin-layer chromatography (TLC) and [α^32^P]-ATP as described [21,27]. Reactions were performed at 37 °C using a fixed DisA concentration (25 nM) and increasing RecG concentrations (7 to 30 nM) in buffer E (50 mM Tris-HCl pH 7.5, 50 mM NaCl, 1 mM DTT, 10 mM MgCl_2_, 0.05 mg/mL BSA, 0.1% Triton, 5% glycerol) containing 100 μM ATP (at a ratio of 1:2000 [α^32^P]-ATP:ATP). After 30 min incubation, the reaction was stopped by adding 50 mM EDTA, 2 μL of each reaction were spotted onto 20 × 20 cm TLC polyethyleneimine cellulose plates and run for about 2 h in a TLC chamber containing running buffer F [1:1 (*v*/*v*) 1.5 M KH_2_PO_4_ (pH 3.6) and 70% ammonium sulfate]. Dried TLC plates were analyzed by phosphor imaging and spots were quantified using ImageJ. T-tests were applied to analyze the statistical significance of the data.

## 3. Results

### 3.1. RecG Does Not Compete with DisA for Binding to HJ DNA

Previously, it has been shown that: (i) DisA preferentially binds to branched intermediates than to ssDNA or dsDNA [21]; and (ii) both RecG*_Eco_* and RecG preferentially binds with high affinity to HJ DNA [31,45]. To test whether DisA works in concert with RecG and if both proteins co-exist on the DNA, they were purified and DNA binding assays to HJ-J3 DNA were performed (Figure 1). The integrity of the HJ-J3 DNA substrate was confirmed by assaying its sensitivity to the HJ-resolvase RecU, as described [49].

DisA bound HJ-J3 DNA with an apparent binding constant (K_Dapp_) of 17 ± 1 nM and formed large molecular mass complexes or aggregates that remained trapped in the well (Figure 1A, lanes 2–4) [21]. It is likely that different protein-DNA complexes formed a net trapped in the well. From the X-ray structure, it can be predicted that an open HJ is sandwiched between two DisA molecules, as it was proposed for RuvA [50]. Here, each RuvA-like (HhH) tetrameric domain of the dumbbell-shaped structure of the DisA octamer bound to a HJ DNA molecule, and it is well-positioned to recognize another HJ DNA [21]. RecG bound the HJ-J3 DNA with a K_Dapp_ of 4 ± 0.3 nM, and formed mainly a G-I complex, although a minor slow-moving and diffused complex (G-II) in the presence of saturating RecG concentrations was also observed (Figure 1A, lanes 5–7). Similarly, RecG*_Eco_* also formed a G-II complex with HJ DNA at high protein concentrations and bound DNA with similar K_Dapp_ [31,45].

Since RecG binds HJ-J3 DNA with ~4-fold higher affinity than DisA, we tested whether RecG competes with DisA for DNA binding. The HJ-J3 DNA was incubated with saturating RecG concentrations, and then increasing DisA concentrations were added. In the presence of limiting DisA (6 nM) the large fraction of the preformed RecG-HJ-J3 DNA complexes did not enter the gel (like the DisA-HJ-J3 DNA complexes) (Figure 1A, lane 8 vs. 2). Similar results were observed if both proteins were added simultaneously, or when DisA-HJ-J3 DNA complexes were performed (Figure 1A, lane 8 vs. 11 and 14). It is likely that the presence of RecG facilitates the recruitment of limiting DisA on the RecG-HJ-J3-DisA complexes, independently of the order of protein addition (Figure 1A, lanes 8 to 16), but we cannot predict if both proteins bind to the same HJ molecule. Alternatively, both proteins bind to a short-lived HJ conformer.

To address whether RecG forms a ternary complex with DisA-[γ^32^P]-HJ-J3 DNA, DNase I footprinting assays were performed. As expected, the center of the junction in the HJ-J3 DNA is in its ssDNA form, thus it was insensitive to DNase I attack (denoted as junction in Figure 1B). RecG in the apo (data not shown) or in the ATPγS bound form (RecG·ATPγS) did not reveal a clear DNase I footprinting (Figure 1B, lanes 2–4). This is consistent with previous reports showing that RecG might only interact with the junction region, as documented for RecG*_Eco_* [51], and therefore it cannot be detected in this assay.

Saturating DisA concentrations showed an extended region of protection from DNase I attack (40–45-bp DNA) (Figure 1B, lanes 6–7). Alternatively, DisA unfolded the HJ DNA, making it insensitive to DNase I attack. Surprisingly, in the presence of RecG, the extended DisA footprint was no longer observed. Here, increasing DisA concentrations only protected shorter regions at both sides of the junction (Figure 1B, framed lanes 8–10). From the EMSA data and the footprint assays, we can envision that there is an interplay between both proteins, and that either RecG relocates DisA and brings it towards the junction region, or RecG stabilizes the cognate HJ conformer, with DisA bound to it to form a ternary DisA-HJ-J3 DNA-RecG complex.

### 3.2. RecG on HJ DNA Blocks DisA-Mediated c-di-AMP Synthesis

Previous reports have shown that: (i) in unperturbed exponentially growing cells, DisA forms a static complex on the nucleoid in ∆*recG* cells, where significant amounts of branched intermediates accumulate, but maintains its dynamic movement in wt cells [27]; (ii) the *disA* gene is epistatic to *recG* in response to DNA bulky and non-bulky lesions [20,27]; and (iii) DisA-mediated c-di-AMP synthesis is decreased upon binding to HJ DNA, whereas DisA bound to duplex DNA barely reduces the synthesis of the essential c-di-AMP, which regulates a wide variety of physiological functions [21,22,27]. At least two mutually exclusive conditions for the putative RecG and DisA interactions at the HJ DNA can be envisioned. First, RecG displaces DisA from the HJ-J3 DNA, with subsequent recovery of its DAC activity. Second, RecG binds to and stabilizes the HJ-J3 DNA, and relocates DisA to it, further reducing DisA-mediated c-di-AMP synthesis. To test whether RecG stabilizes a preferred conformer and re-locates DisA to it or if RecG itself affects DisA-mediated c-di-AMP synthesis, the DAC activity of DisA was analyzed.

At 5 mM ATP, the maximal unwinding activity of RecG was observed [45]. However, steady-state kinetic analysis of DisA (25 nM) revealed that it efficiently converted two ATP molecules into c-di-AMP in the presence of 10 mM Mg^2+^, and its DAC activity was saturated at ~300 μM ATP (Appendix A). Thus, we have used conditions (100 μM ATP) in which DisA maintains its DAC activity, but RecG-mediated unwinding of HJ DNA is poorly detected. Indeed, after 30 min incubation, RecG (30 nM) did not catalyze c-di-AMP synthesis (Figure 1C, lane 8). It is worth mentioning that under the conditions used to detect [α^32^P]-c-di-AMP, [α^32^P]-ADP is poorly separated from the [α^32^P]-ATP substrate, therefore, RecG-mediated hydrolysis of ATP cannot be observed [27]. In the presence of 100 μM ATP (at a ratio of 1:2000 [α^32^P]-ATP:ATP), DisA (25 nM) efficiently synthesized c-di-AMP, with ~75% of the radiolabeled ATP converted to c-di-AMP. Nearly equimolecular concentrations of RecG (30 nM) did not significantly affect the DAC activity of DisA (Figure 1C, lane 2 vs. 7).

Addition of HJ-J3 DNA (125 nM), at a 5:1 stoichiometry relative to DisA, reduced c-di-AMP synthesis (~2-fold, *p* < 0.01) (Figure 1C, lane 3 vs. 2), as earlier reported [21,27]. In the presence of HJ-J3 DNA, limiting RecG concentrations (7 nM, a RecG:DisA molar ratio of 0.3:1) significantly blocked the DAC activity of DisA by ~11-fold (*p* < 0.01) (Figure 1C, lane 4). This suggests that RecG, under these experimental conditions that do not allow DNA unwinding, further inhibits DisA-mediated c-di-AMP synthesis. Higher RecG concentrations did not reverse such inhibition (Figure 1C, lanes 5–6). It is likely that RecG bound to the HJ-J3 DNA relocates DisA to a position in which DisA-mediated c-di-AMP synthesis is blocked, rather than competing or displacing DisA from the HJ DNA, displacement that would counteract the negative effect of HJ DNA on DisA-mediated c-di-AMP synthesis. We interpreted this observation as that RecG contributes to limit c-di-AMP synthesis, and this effect indirectly blocks cell proliferation. This is consistent with the observation that: (i) in response to MMS-induced DNA lesions, the c-di-AMP levels decreased to levels comparable to that in the absence of DisA [26]; (ii) low c-di-AMP levels indirectly increase (p)ppGpp synthesis [52,53]; and (iii) an increase of pppGpp levels inhibits DNA replication by inhibiting DNA primase activity [54].

### 3.3. DisA Does Not Form a Stable Complex with RecG

In unperturbed exponentially growing cells, both DisA (~75 octamers/colony forming unit [CFU]) and RecG (~40 RecG monomers/CFU) are low abundant proteins [27,55]. Thus, to gain insight into a putative DisA interaction with RecG, a bacterial two-hybrid assay was used (see Materials and Methods). The 5´- or 3´-end of the genes was fused to the coding sequence of either the T18 or T25 domain of the *Bordetella* adenylate cyclase. If the fused pair interacts, it allows the interaction of these domains, and the enzyme is reconstituted. Then, cAMP is synthesized, and the cAMP-bound catabolite activator protein becomes activated. This induces the expression of the *lacZ* gene and β-galactosidase activity. Unexpectedly, full-length RecG interacted with itself rendering blue colored colonies. RecG failed to render blue colonies when combined with DisA (and *vice versa*), suggesting that no stable interaction between these proteins exist (Appendix A).

To confirm this result, we used a His-tagged DisA (1 µg) variant and native RecG (0.5 µg). DisA (predicted mass of 40.7 kDa) is contaminated with traces of His-DisA-bound to c-di-AMP (a complex that runs as a 41 kDa protein). His-DisA was retained in the 50-µl Ni^2+^ matrix through coordination with its C-terminal histidine tag, and eluted (E) with 400 mM imidazole (Appendix A, lane 6). However, apo RecG (predicted mass 78.1 kDa) was mainly present in the flow-through (FT), and the traces of the protein entrapped in the Ni^2+^ matrix were released after the first wash (W1) (Appendix A, lanes 7–8). When both His-tagged DisA and RecG were loaded into the Ni^2+^ column, we observed that Apo RecG was present in the FT (Appendix A, lane 3) while DisA was observed in the E fraction in the presence of 400 mM imidazole (Appendix A, lane 6). This suggests that DisA does not form a stable complex with RecG, at least in a way that can be detected using these assays (Appendix A).

### 3.4. RecG Is Mechanistically Suited for Fork Reversal and Fork Restoration

Previously it has been shown that: (i) both RecG*_Eco_* and RecG preferentially bind branched structures, such as HJs [31,45,56]; and (ii) RecG*_Eco_* preferentially binds to stalled forks and catalyzes the formation of a HJ-like DNA structure (fork reversal), and upon binding to a reversed fork catalyzes its migration in the opposite direction to yield a restored fork [10,11,30,34]. To test whether RecG works in a similar fashion to the well-characterized RecG*_Eco_* protein, different substrates were tested. Except the HJ-J4 DNA, the substrates have heterologous arms to prevent their spontaneous branch migration. HJ-J4 contains a central 12-bp homologous sequence so it can spontaneously migrate.

In the presence of ATP (5 mM), RecG (15 to 30 nM) remodeled a HJ, a 3´-fork (a replication fork with a fully synthesized leading-strand and no synthesis in the lagging-strand), and a 5´-fork (a replication fork with a fully synthesized lagging-strand and no synthesis in the leading-strand) with apparent similar efficiency, yielding a flayed DNA product (an unreplicated fork) that was not further processed (Figure 2, lanes 1–3 and 7–12). Indeed, a flayed DNA substrate was not unwound by RecG to release the free labelled ssDNA strand (Figure 2, lanes 13–15). When a Y-fork (a replication fork with fully synthesized leading- and lagging-strands) with no complementary arms was tested (Figure 2, lanes 4–6), RecG mediated fork reversal, yielding a flapped and the radiolabeled nascent leading-strand, because the nascent strands are heterologous and cannot anneal, to avoid spontaneous branch migration (Figure 2, lanes 5–6). The indicated substrates, intermediates or products were run in parallel (Figure 2, lanes 1, 4, 7, 10, 13, 15 and 16) to infer their mobility.

From these results, we conclude that: (i) RecG, as RecG*_Eco_* [10,30,34], may catalyze both fork regression and fork restoration (Figure 2, lanes 1-12); and (ii) RecG may process partially replicated forks (3´- and 5´forks), which are isomers of a D-loop, with similar efficiency. However, not all activities associated with RecG*_Eco_* are observed with RecG. For example, RecG*_Eco_* unwinds R-loop structures, but this activity is not seen with RecG [40].

### 3.5. DisA Limits RecG-Mediated Fork Reversal

Previously, it has been shown that: (i) DisA and RecG bind with significantly high affinity branched structures (e.g., HJ DNA) [21,45]; and (ii) RecG*_Eco_* binds to stalled forks and preferentially reverses the one with a leading-strand gap [10,11,30,34]. To test whether RecG shows a preference for a stalled fork with a leading-strand gap, and if DisA affects RecG-mediated fork remodeling, synthetic replicated forks with a 15-nt ssDNA gap on the parental leading-strand (forked-Lead) or on the parental lagging-strand gap (forked-Lag) were constructed.

As revealed in Figure 3A,C lanes 1, 2 and 13, the indicated forked-Lead or forked-Lag substrates, intermediates or products were run to infer their mobility. The substrates have heterologous arms to prevent spontaneous branch migration. Due to that, the displaced nascent strands cannot rewind to duplex DNA. Nevertheless, we can visualize the displacement of the nascent strands and the accumulation of flayed DNA (cartoons in Figure 3A,C).

In the presence of ATP (5 mM), DisA did not reverse a forked-Lag or forked-Lead DNA substrate (Figure 3A,C, lanes 3–5), whereas RecG (15 nM) reversed both in the absence of any accessory protein, yielding a flayed product with complete unwinding of the nascent strands from the template (Figure 3A,C, lane 6). RecG exhibited a very slight preference for the unwinding of the forked-Lead over the forked-Lag substrate (Figure 3A,C, lane 6). The RecG*_Eco_* translocase, however, shows a clear preference to reverse a forked-Lead DNA substrate [10,30,34].

When a fixed RecG concentration was pre-incubated with the forked-Lead or forked-Lag substrate, and then ATP and increasing DisA concentrations were added, RecG-mediated fork reversal was only slightly affected even at high DisA:RecG (3.2:1) ratios (~1.5-fold, *p* > 0.1) (Figure 3A,C, lanes 7–9). Conversely, when increasing DisA concentrations were pre-incubated with the forked-Lead or forked-Lag substrate, and then RecG and ATP were added, RecG-mediated fork reversal was strongly inhibited (*p* < 0.01) (Figure 3A,C, lanes 10–12). At about stoichiometric concentrations (1:1.2 DisA:RecG ratios), RecG-mediated fork reversal was inhibited by ~4-fold (Figure 3B,D). The competitive nature of the DisA effect on RecG-mediated fork reversal was inferred by the order of protein addition and the relative stoichiometry of the reaction (Figure 3A–D). This suggests that if DisA is bound to the stalled replication fork, it inhibits the unwinding by RecG (Figure 3B,D). Alternatively, DisA binds to the fork structure and unspecifically (as a roadblock) inhibits RecG-mediated fork reversal. However, we consider this hypothesis unlikely, because DisA neither affects PriA-dependent replication initiation nor DNA replication elongation [20].

### 3.6. DisA Inhibits RecG-Mediated Fork Restoration

A HJ structure is processed by RecG, leading to a resetting of the nascent strands back to their original configuration (fork restoration) (Figure 2, lanes 2–3), as earlier described for RecG*_Eco_* [10,11,34,56]. Thus, we tested whether this activity is likewise affected by DisA.

In the presence of ATP (5 mM), DisA did not restore a reversed fork (Figure 3E, lanes 9–11), whereas RecG (15 nM) catalyzed dissociation of ~73% of a mobile HJ structure (HJ-J4 DNA) to produce two flayed duplexes that co-migrate (Figure 3E, lane 1).

When RecG was pre-incubated with HJ-J4 DNA, and then a limiting DisA concentration and ATP were added, the dissociation of HJ-J4 DNA was slightly, but not significantly, increased (*p* > 0.1) (Figure 3E, lane 2). When higher DisA concentrations were used, the dissociation of HJ-J4 DNA was only slightly reduced, ~1.3-fold even at a 3.2:1 DisA:RecG ratios (Figure 3E, lanes 3–4 and 3F grey bars). Conversely, when the HJ-J4 DNA was pre-incubated with increasing DisA concentrations and then RecG and ATP were added, the dissociating activity of HJ DNA was reduced at about stoichiometric concentrations, but strongly inhibited at a 3.2:1 DisA:RecG ratios (*p* < 0.01) (Figure 3E, lanes 5–7 and 3F empty bars). We can envision that DisA binds to the reversed replication fork and counteracts the unwinding by the RecG translocase. Alternatively, DisA binds to stalled or reversed forks and impedes strand separation by any helicase, acting as a strong protein block.

To test whether DisA bound to stalled or reversed forks impedes strand separation by any other DNA helicase [see 57], the PcrA DNA helicase was tested. Since PcrA does not branch migrate HJ structures (B.C. unpublished results), we extended the nascent leading-strand (3´ tail HJ DNA substrate) or lagging-strand (5´-tail HJ DNA) of the HJ to provide an entry site for PcrA (Appendix A). As previously described, PcrA preferentially unwinds a 3´tailed DNA substrate [57]. In the presence of increasing DisA concentrations (12.5 to 100 nM), the helicase activity of PcrA (15 nM) over both DNA substrates did not significantly vary (*p* > 0.1) even at a 6.6:1 DisA:PcrA stoichiometry (Appendix A). Since at these DisA concentrations all the HJ DNA should be bound by DisA (see Figure 1B), but this does not inhibit PcrA unwinding activity, it is likely that DisA inhibition of RecG processing of reversed forks is specific.

### 3.7. DisA Inhibits the DNA-Dependent ATPase Activity of RecG

To understand how DisA may regulate RecG helicase activity, kinetic studies of RecG-mediated ATP hydrolysis in the presence of DisA were conducted via an ATP/NADH-coupled spectrophotometric enzymatic assay. In the absence of DNA, the ATPase activity of RecG (15 nM) was undetectable, with a catalytic constant (K_cat_) of <0.1 min^−1^ (Figure 4A, light blue line, Table 1). In the presence of limiting protein concentrations (5 nM, 1 monomer of RecG/ 2000-nt), RecG hydrolyzed ATP with higher efficiency upon binding to circular ssDNA (cssDNA) than to HJ DNA (Figure 4A, blue and green lines). Under the conditions used (10 mM MgCl_2_), the cssDNA is folded onto secondary structures that mimic flayed structures. Similar results were reported for RecG*_Eco_* [33,51,58]. The saturation of the maximal rate of ATP hydrolysis by RecG was clearly observed in our assays due to the strong ATPase activity of RecG and the consumption of some reactants. The maximal rate of ATP hydrolysis showed a continuous increase by doubling the concentration of the reactants (Figure 4A, insert, dark blue line).

DisA (36 nM) did not show any detectable hydrolysis of ATP in the presence of HJ or cssDNA (Figure 4B,C, purple line, Table 1), because hydrolysis of the two ATP molecules to form c-di-AMP cannot be measured with this ATP/NADH-coupled assay since the DAC reaction releases PPi instead of Pi. Furthermore, in the presence of an excess of ATP (5 mM), the DAC activity of DisA was reduced [42].

When HJ DNA was simultaneously incubated with RecG and DisA, RecG-mediated ATP hydrolysis was not affected at about stoichiometric RecG:DisA concentrations (molar ratio of 1:1.6) (Figure 4B, brown vs. dark blue line, Table 1). However, ATP hydrolysis was significantly reduced at a RecG:DisA 1:2.4 ratio (*p* < 0.01) (Figure 4B, red vs. dark blue line, Table 1). The mean ATPase activity expressed as K_cat_ values ± SEM, estimated from the slope of the curves, can be found in Table 1.

With cssDNA as effector, RecG (15 nM, 1 RecG monomer/666-nt) catalyzed ATP hydrolysis at the maximum rate of ~4000 min^−1^ (Figure 4C, dark blue line, Table 1). When DisA was simultaneously included in the reaction, RecG-mediated ATP hydrolysis was significantly inhibited (~4- and ~9-fold, *p* < 0.01) at about stoichiometric concentrations (RecG:DisA molar ratios of 1:0.8 and 1:1.6, respectively) (Figure 4C, green and brown lines, Table 1). A RecG:DisA molar ratio of 1:2.4 strongly inhibited RecG-mediated ATP hydrolysis by ~16-fold (*p* < 0.01) (Figure 4C, red line, Table 1).

Then, it seems that DisA regulates RecG-mediated ATP hydrolysis. Alternatively, DisA forms large complexes on the cssDNA that unspecifically inhibit the ATPase activity of RecG. To test whether DisA-mediated inhibition of RecG is specific, we analyzed the ATPase activity of PcrA (Appendix A). In the presence of low (1 DisA/100-nt) or high DisA (1 DisA/12-nt) concentrations, the ATPase activity of PcrA (15 nM, 1 PcrA/ 660-nt) did not significantly vary (k_cat_ of 1722 ± 332 and 1736 ± 297 min^−1^ vs. 1750 ± 382 min^−1^, *p* > 0.1) (Appendix A, green and red vs. blue line). This confirms that the inhibition of the ATPase activity of RecG by DisA is a genuine and specific activity of DisA.

### 3.8. DisA Competes with RecG for Binding to the cssDNA Substrate

The helicase assays showed that the inhibitory effect of DisA is higher when pre-bound to the DNA. We can envision that DisA bound to the cssDNA substrate counteracts the loading of RecG and thereby inhibits RecG-mediated ATP hydrolysis. To test this hypothesis, the effect of order of protein addition was tested. When RecG was pre-incubated with cssDNA (RecG + cssDNA) (5 min at 37 °C), and then DisA (24 nM, 1 octamer DisA/416-nt) and ATP were added, ATP hydrolysis was almost not affected (Figure 4D brown vs. dark blue line, Table 1). By contrast, when DisA pre-bound the cssDNA (DisA + cssDNA) (5 min at 37 °C), and then RecG and ATP were added, ATP hydrolysis was significantly inhibited by ~9-fold (*p* < 0.01) (Figure 4D, red vs. dark blue line, Table 1), as when both proteins were simultaneously added (Figure 4C, brown line). This suggests that DisA binds cssDNA (or formed secondary structures [e.g., flayed structures]) and counteracts RecG loading, inhibiting RecG-mediated ATP hydrolysis. Since the inhibition of the RecG ATPase occurs in the presence of limiting DisA (1 DisA/416-nt) we have to assume that DisA might bind to discrete sites in the cssDNA, probably regions with secondary structures, which are also the binding site for RecG.

To confirm that DisA competes with RecG for cssDNA binding, DisA was replaced by DisA ∆C290, a mutant variant which lacks the C-terminal DNA-binding domain [24]. The presence of a saturating DisA ∆C290 concentration (36 nM, 1 DisA/277-nt) did not inhibit RecG-mediated ATP hydrolysis (Figure 4E, Table 1). The same was observed when HJ DNA was used as the effector (Figure 4F, Table 1).

DisA could also inhibit the ATPase activity of RecG by the consumption of the ATP substrate through its conversion to c-di-AMP. Moreover, c-di-AMP itself could inhibit the ATPase activity of RecG. This is probably not the case because c-di-AMP synthesis is inhibited when DisA is bound to HJ DNA (Figure 1C) and DisA ∆C290 does not inhibit RecG but still synthesizes c-di-AMP [24]. To further discard these hypotheses, DisA was replaced by DisA D77N, a mutant variant with a point mutation in the DAC active site that does not synthesize c-di-AMP [21,24]. In the presence of cssDNA, RecG-mediated ATP hydrolysis was inhibited in a DisA D77N concentration-dependent manner, to a similar extent as when wt DisA was used (Figure 4G, Table 1). As shown for wt DisA, higher DisA D77N concentrations were required to observe a similar rate of inhibition of ATP hydrolysis in the presence of HJ DNA (Figure 4H, Table 1). This proves that the inhibition of RecG-mediated ATP hydrolysis is a genuine activity associated with DisA bound to cssDNA or HJ DNA, and independent of c-di-AMP and of a reduction of the ATP pool.

## 4. Discussion

Our work provides new insights into a comprehensive model of DisA and RecG functions contributing to genome integrity by stabilizing stalled or reversed replication forks. In *E. coli*, four proteins (RecA, RecG, RuvAB and RecQ) may process stalled or reversed forks [9,10,11,59]. However, when the single genome of an inert mature haploid *B. subtilis* spore is damaged, unperturbed spore revival requires RecA, RecG and RuvAB, but not a RecQ-like DNA helicase (RecS or RecQ) [17,20]. Furthermore, although it is poorly known if these enzymes act in coordination with each other, the *disA* gene is epistatic to the *recA*, *recG* or *ruvAB* gene in response to DNA damage [20]. The role of DisA in limiting RecA -mediated DNA strand exchange was previously reported [43]. Here, we report the interplay of DisA with RecG, and how RuvAB and DisA might contribute to replication fork remodeling and replication restart will be addressed elsewhere.

We are aware that the phenotypes seen in cells lacking RecG are very complex [29,60], and that DisA only contributes to the repair of perturbed replication forks [20,27]. In fact, the inactivation of *B. subtilis recG* renders cells extremely sensitive to DNA damaging agents that promote single-strand gaps and DSBs [60], while the absence of DisA renders cells only sensitive to single-strand gaps, but they remain recombination proficient and are apparently as capable of repairing DSBs as the wt control [26,27], suggesting that DisA might only react to replication stress. However, it seems that although RecG may have other functions in which DisA is not involved, its roles upon replication stress are controlled by DisA.

RecG*_Eco_* and RecG are evolutionarily separated by more than 1500 million years. *In vitro*, both enzymes, as well as mammalian SMARCAL1, unwind a stalled fork to form a HJ-like structure (fork reversal) and regress a HJ DNA leading to fork restoration (Figure 2 and Figure 3) [10,11,30,34,56]. Thus, it is likely that RecG-mediated replication fork remodeling is one of its genuine in vivo activities, well conserved in evolution. In this work, we observed that DisA bound to the branched intermediate limits both RecG activities at stalled and reversed forks (Figure 3). The effect of DisA seems to be specific because DisA neither affects PriA-dependent replication initiation, nor DNA replication elongation [20]. To confirm this observation, we have shown here that DisA neither affects PcrA-mediated ATP hydrolysis nor unwinding of tailed HJ DNA substrates (Appendix A), suggesting that DisA does not act as a protein roadblock that inhibits RecG activities.

DisA monitors genomic stability by forming a rapid-moving focus on the nucleoid [23]. Upon perturbation of DNA replication, DisA pauses in a RecO- and RecA-dependent manner, perhaps at stalled replication forks [24]. This is consistent with the observation that, upon UV treatment, >80% of the RecA foci co-localized with the replisome [61] rather than with the lesion-containing gap left behind by the replisome. Here, RecA interacts with and loads DisA at branched DNA (stalled or reversed forks), as observed in vitro [24]. DisA bound to HJ DNA drops the synthesis of c-di-AMP and this inhibition is additive in the presence of RecG (Figure 1C), although these proteins do not form a stable complex (Appendix A). These results suggest that DisA transiently interacts with RecG at a HJ intermediate, with a subsequent blockage of DisA-mediated c-di-AMP synthesis to provide a fail-safe mechanism that avoids the uncoupling of the cell cycle. Low c-di-AMP levels indirectly increase the production of (p)ppGpp [52,53], that in turn inhibits DNA primase [54], blocking cell proliferation (Figure 5).

Our results suggest that DisA, which is absent in *E. coli* but present in the Firmicutes, Actinobacteria, Fusobacteria and Termotogae phyla, is implicated in controlling the processing of a perturbed replication fork and contributes to genome integrity by stabilizing stalled or reversed replication forks. Why is relevant to limit fork processing? It seems that repair-by-recombination of a stalled fork is a two-edged sword. If a damaged template base stalls the replisome, and then the stalled fork is reversed, the RuvAB-RecU complex could catalyze HJ cleavage (Figure 5A,B). This would lead to the formation of a one-ended DSB, that should be lethal if an intact homologous template is absent and end resection function have not been synthesized yet, as at the early period of spore outgrowth [15,16]. We propose that by limiting RecG-mediated fork reversal, DisA provides the timeframe to facilitate the removal of the DNA lesion. We show, however, that if RecG is pre-bound to a stalled or reversed fork, it becomes insensitive to DisA action (Figure 3), suggesting that DisA competes with RecG for binding to stalled or reversed forks.

Earlier, we have proposed that DisA also limits RecA-mediated DNA strand exchange and template switching (Figure 5C) [24]. By doing so, DisA might facilitate that RecA protects the extruded strands from degradation, keeping chromosomal stability. Similarly, eukaryotic Rad51 promotes replication fork reversal and protects the extruded nascent strands from degradation [62]. In addition, Rad51 mediators and/or Rad51 itself in eukaryotes modulate fork stability by limiting SMARCAL1 [63,64], in a way similar as here DisA does with RecG. This way, DisA might contribute to the election of the DDT sub-pathway (translesion synthesis, template switching, fork reversal) to overcome replicative stress (Figure 5C).

In summary, we illustrate how the DisA checkpoint may help to direct the flow of protein-DNA intermediates from a stalled fork to replication restart preventing fork nucleolytic degradation (Figure 5). Furthermore, by doing so, DisA might also help to control low error rate DDT pathways to maintain the balance between genetic stability and diversity. Some key questions that remain open for future studies will help us to define which branch migration translocase is involved in replication restart in *B. subtilis*. First, it will relevant to determine whether RecA, and perhaps DisA and RecG, contributes to the recruitment of the PriA preprimosomal protein at stalled forks, leading to the resumption of DNA replication. Finally, it will be relevant to determine how DisA coordinates other fork remodeling enzymes (e.g., RuvAB); and determine the interplay of RecG with other DNA helicases at replication-transcription conflicts. The presence of DisA also in non-spore forming bacteria indicates that it has a broader role in genome stability than just during spore revival.

## Figures and Tables

**Figure 1 cells-10-01357-f001:**
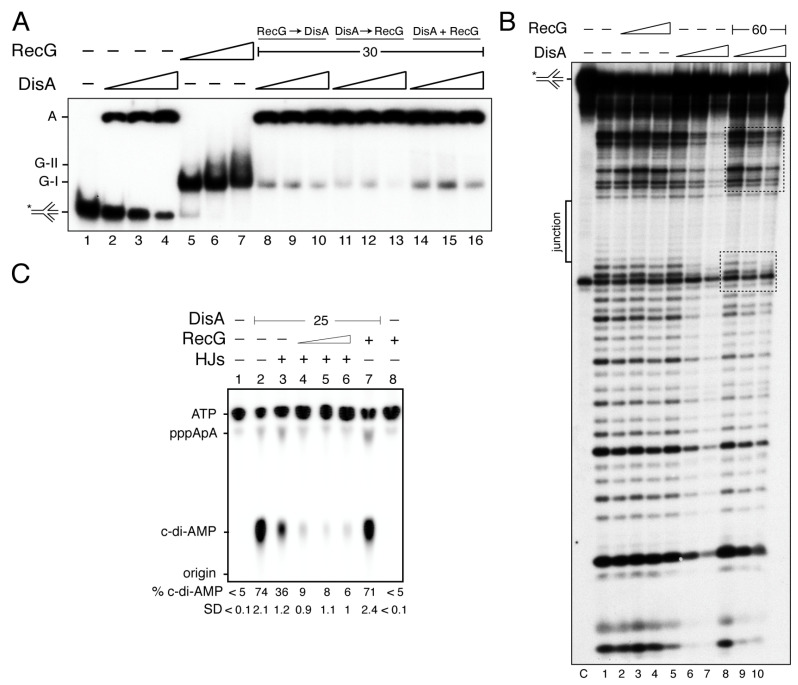
DisA may coexist with RecG on HJ DNA. (**A**) For EMSA assays [γ^32^P] HJ-J3 DNA (0.1 nM in molecules) was incubated with increasing DisA (6–25 nM, lanes 2–4), RecG (7.5–30 nM, lanes 5–7) or increasing DisA and fixed RecG (lanes 14 to 16) concentrations, or pre-incubated with a fixed RecG (lanes 8–10) or increasing DisA concentrations (lanes 11–13) and then the second protein added for 15 min in buffer B containing 5 mM ATPγS at 37 °C. The protein-DNA complexes were separated using 6% PAGE in TAE buffer and visualized by autoradiography. The RecG-HJ (G-I and G-II) and DisA-HJ (A, retained in the well) complexes are indicated. (**B**) DNAse I footprinting assays. The [γ^32^P] HJ-J3 DNA was pre-incubated with increasing RecG (15–60 nM) or DisA concentrations (30–120 nM) or increasing DisA concentrations and a fixed RecG concentration (60 nM) in buffer C containing 5 mM ATPγS for 15 min at 37 °C, and then DNAse I was added. Samples were separated in 15% dPAGE and autoradiographed. The region protected by DisA in the presence of RecG is marked with rectangles. C denotes a HJ DNA control without DNase I treatment. In (A,B) the assay was repeated >3 times with similar results, a representative gel is shown. (C) RecG bound to HJ DNA inhibits DisA-mediated c-di-AMP synthesis. DisA (25 nM) was incubated in buffer E containing 100 µM ATP in the presence of HJ-J3 DNA (125 nM) and increasing RecG concentrations (7 to 30 nM**)** for 30 min at 37 °C. Samples were separated by TLC, dried and visualized by phosphor imaging. Spots corresponding to the ATP and c-di-AMP molecules were quantified. The position of ATP, the linear pppA-pA intermediate, c-di-dAMP and the origin are indicated. A representative gel and the mean ± SD of >3 independent experiments are shown. The radiolabeled strand is indicated with an asterisk.

**Figure 2 cells-10-01357-f002:**
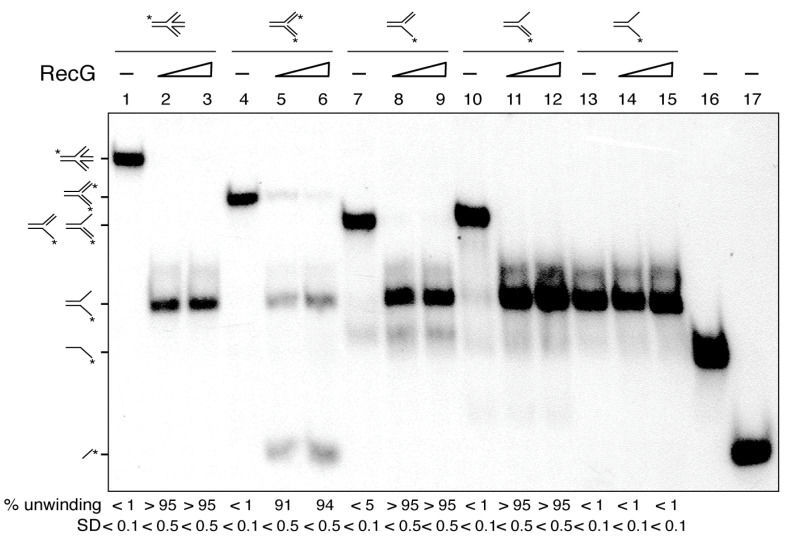
RecG unwinds HJs and nascent DNA strands from partial and complete replication fork substrates. The different radiolabeled DNA substrates (HJ-J4, Y fork, 5´-fork, 3´-fork or flayed DNA) were incubated with increasing RecG (15 and 30 nM) concentrations in buffer C containing 5 mM ATP (15 min at 37 °C). After deproteinization, substrates and products were separated by 6% PAGE in TAE buffer and analyzed by auto-radiography. The radiolabeled strands are indicated with an asterisk. In lanes 16 and 17 the [γ^32^P]-labelled oligonucleotides are loaded as controls. A representative gel and below the mean ± SD of 3 independent experiments is shown.

**Figure 3 cells-10-01357-f003:**
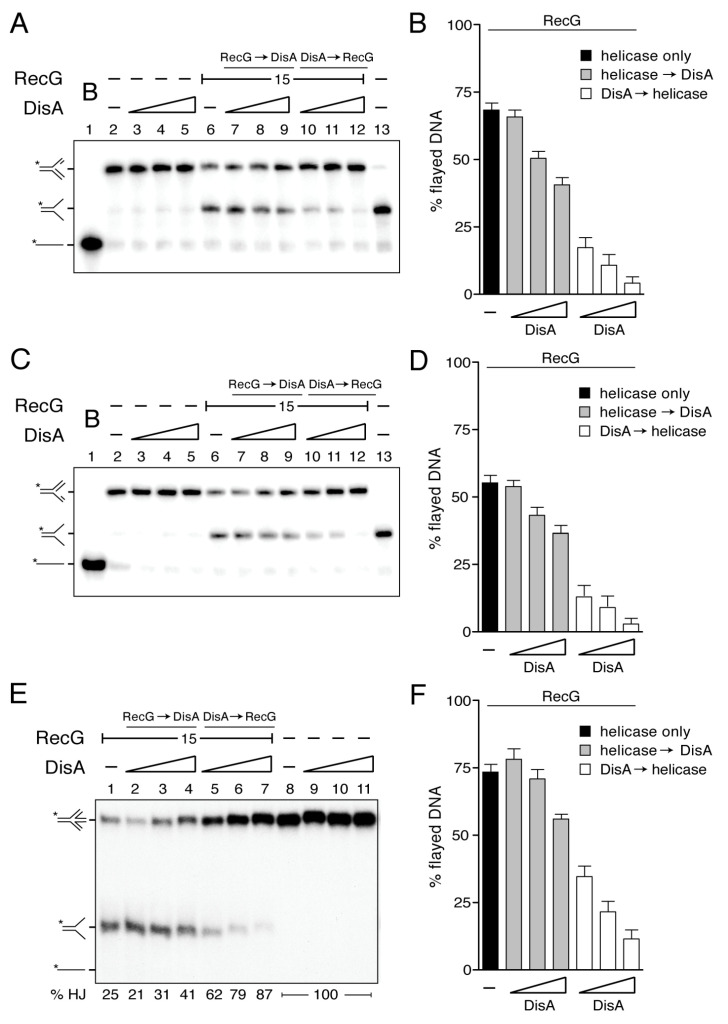
DisA inhibits RecG-mediated fork reversal and HJ regression. (**A**,**C**,**E**) [γ^32^P] forked-Lead (**A**), [γ^32^P] forked-Lag DNA (**C**), or [γ^32^P] HJ-J4 DNA (0.1 nM) (**E**) was pre-incubated with increasing DisA concentrations (doubling from 12–48 nM) or a fixed amount of RecG (15 nM) in buffer B containing 10 mM MgCl_2_ (5 min at 37 °C). Then, the second protein (variable DisA [RecG → DisA], a constant amount of RecG [DisA → RecG] or no protein) and 5 mM ATP were added and the reaction was further incubated for 15 min at 37 °C. The reaction was stopped, deproteinized and separated by 6% native PAGE. Gels were dried and visualized by phosphor imaging. **-,** no protein added; B, boiled product. (**B**,**D**,**F**) The relative amount of flayed DNA product in three independent experiments such as those shown in A, C and E was quantified, and the mean ± SD are represented. The radiolabeled strand is indicated with an asterisk.

**Figure 4 cells-10-01357-f004:**
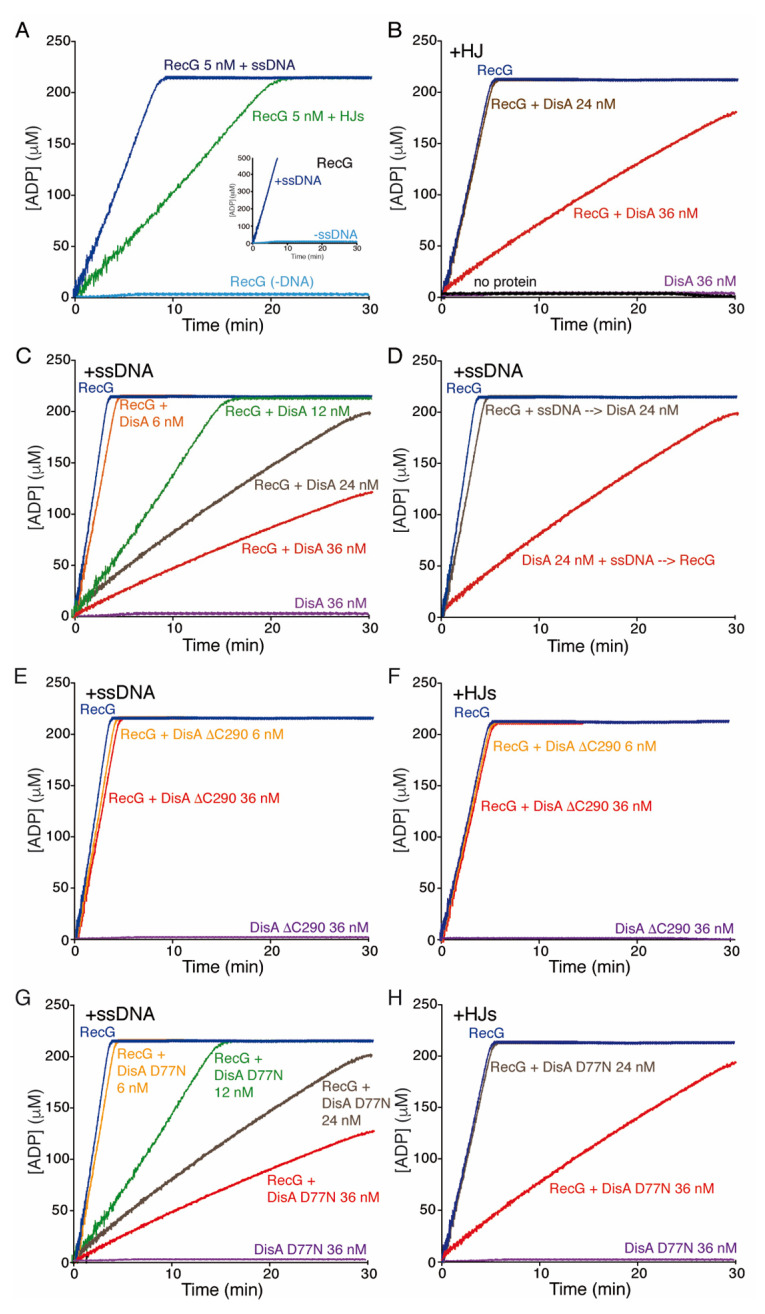
DisA inhibits the ATPase activity of RecG. (**A**) RecG (5 nM) was incubated with or without 3199-nt cssDNA or HJ-3 DNA (10 μM in nt) in buffer D containing 5 mM ATP and the ATP regeneration system, and the ATPase activity was measured for 30 min. Insert, RecG (15 nM) was incubated with cssDNA in buffer D containing 5 mM ATP and 2x the ATP regeneration system, and the ATPase activity was measured for 30 min. (**B**) HJ-J3 DNA was incubated with RecG (15 nM), DisA (24–36 nM) or both in buffer D containing 5 mM ATP and the ATP regeneration system and the ATPase activity measured for 30 min. (**C**) cssDNA was incubated with RecG (15 nM), DisA (6–36 nM) or both proteins and the ATPase activity was measured. (**D**) cssDNA was pre-incubated with RecG (15 nM) or DisA (24 nM) (5 min at 37 °C), then DisA (RecG + ssDNA → DisA) or RecG (DisA + ssDNA → RecG) was added and the ATPase activity was measured. (**E** to **H**) cssDNA (**E**,**G**) or HJ-3 DNA (**F**,**H**) was incubated with RecG (15 nM) and DisA ∆C290 (6–36 nM) (**E**,**F**) or with RecG (15 nM) and DisA D77N (6–36 nM) (**G**,**H**) and the ATPase activity was measured. All reactions were repeated three or more times with similar results, and a representative graph is shown here.

**Figure 5 cells-10-01357-f005:**
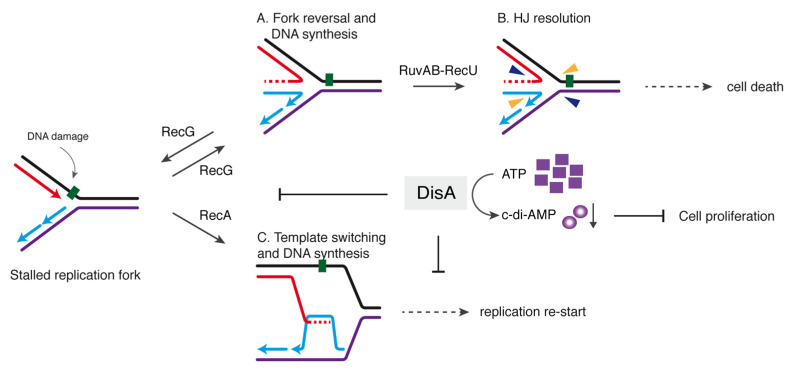
Model for the control of fork remodeling by DisA. DisA moves onto dsDNA synthesizing c-di-AMP until an unrepaired DNA lesion (represented here by the green square on the leading strand template) causes blockage of replication fork movement. The fork remodeler RecG (or RecA) helps DisA to bind the stalled or reversed fork and then DisA-mediated c-di-AMP synthesis is suppressed. Low levels of c-di-AMP indirectly increase in turn (p)ppGpp synthesis which that inhibits DNA replication. In the upper panel, DisA suppresses RecG-mediated fork reversal to avoid fork breakage (**A**). If fork reversal occurs, RecU in concert with RuvAB may catalyze HJ cleavage (**B**), leading to the formation of a one-ended DSB that is lethal during spore revival. In the lower panel, (**C**) RecA-mediated template switching followed by DNA synthesis is also modulated by DisA to allow time for specific enzymes to repair the damage, and then replication re-starts.

**Table 1 cells-10-01357-t001:** RecG rates of DNA-dependent ATP hydrolysis.

DNA Substrate and Proteins	K_cat_ (min^−1^) ^a^
RecG (15 nM), no ssDNA	<0.1
RecG + HJ DNA	2777 ± 47
RecG + HJ DNA + 24 nM DisA^d^	2770 ± 45
RecG + HJ DNA + 36 nM DisA^e^	370 ± 10
HJs + 36 nM DisA	<0.1
RecG + HJ DNA + 24 nM DisA D77N^e^	2772 ± 51
RecG + HJ DNA + 36 nM DisA D77N^e^	403 ± 17
HJs + 36 nM DisA D77N	<0.1
RecG + HJ DNA + 24 nM DisA D77N^e^	2781 ± 42
RecG + HJ DNA + 36 nM DisA D77N^e^	2745 ± 37
HJs + 36 nM DisA D77N	<0.1
RecG + ssDNA	4000 ± 51
RecG + ssDNA + 6 nM DisA^b^	3333 ± 43
RecG + ssDNA + 12 nM DisA^c^	952 ± 14
RecG + ssDNA + 24 nM DisA^d^	435 ± 9
RecG + ssDNA + 36 nM DisA^e^	256 ± 4
ssDNA + 36 nM DisA	<0.1
RecG + ssDNA → 24 nM DisA^d^	3234 ± 39
24 nM DisA^d^ + ssDNA → RecG	425 ± 11
RecG + ssDNA + 6 nM DisA ∆C290^b^	3991 ± 37
RecG + ssDNA + 12 nM DisA ∆C290^c^	3993 ± 39
RecG + ssDNA + 24 nM DisA ∆C290^d^	3981 ± 45
RecG + ssDNA + 36 nM DisA ∆C290^d^	3970 ± 36
ssDNA + 36 nM DisA ∆C290^d^	<0.1
RecG + ssDNA + 6 nM DisA D77N^b^	3547 ± 35
RecG + ssDNA + 12 nM DisA D77N^c^	1066 ± 29
RecG + ssDNA + 24 nM DisA D77N^d^	467 ± 13
RecG + ssDNA + 36 nM DisA D77N^d^	467 ± 13
ssDNA + 36 nM DisA D77N^d^	<0.1

^a^ ATP hydrolysis was measured as indicated in Materials and Methods. The protein(s) were pre-incubated with ssDNA or HJ DNA (10 µM in nt). The stoichiometry of DisA with ssDNA is indicated (DisA/200^b^-, 100^c^-, 50^d^- or 33^e^-nt). The steady-state kinetic parameters for RecG (1 protein/ 667-nt) were derived from the data presented in Figure 4. The average rate of ATP hydrolysis was obtained from more than three independent experiments and it is shown as the mean ± SEM.

## Data Availability

Datasets were generated during the study. We endorsed MDPI Research Data Policies.

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
