# Peer review of "DisA Limits RecG Activities at Stalled or Reversed Replication Forks"

_cells, 2021, doi:10.3390/cells10061357_

Round 1

Reviewer 1 Report

Torres et al show a detailed biochemical study of the interaction and possible role of DisA in conjuction with RecG in Bacillus subtilis. Experiments are well performed and controlled. Authors show that even thought they could not show direct interaction between the two proteins their activities are mutually affected. The study shed new light into the role of DisA and open new questions in the field.

Minor things:

Page6 Line 246: correct ‘assys’ for ‘assays’

In Figure1 a), is A indicating the position of the wells?; and in b), what is letter ‘C’ indicating?

Page 6 line 257 here and later the authors refer to the Kdapp. The text will benefit if a brief description in the methods, or as a supplementary figure showing the data and analytical method used for estimating the constants, is included.

Figure 4 labels will be more legible with bigger fonts.

page 13 line 529 remove extra dots from: vs…

page 14 line 556 remove extra dots from: vs…

page 16 line 645 reference format seems wrong

page 16 line 663 correct to ‘will be relevant’

Author Response

The referee’s comments are shown in CAPITAL LETTERS  and the answers to their queries in small case letters. The changes are written in a red font in Torres et al wCorrections

REVIEWER 1

TORRES ET AL SHOW A DETAILED BIOCHEMICAL STUDY OF THE INTERACTION AND POSSIBLE ROLE OF DISA IN CONJUCTION WITH RECG IN BACILLUS SUBTILIS. EXPERIMENTS ARE WELL PERFORMED AND CONTROLLED. AUTHORS SHOW THAT EVEN THOUGHT THEY COULD NOT SHOW DIRECT INTERACTION BETWEEN THE TWO PROTEINS THEIR ACTIVITIES ARE MUTUALLY AFFECTED. THE STUDY SHED NEW LIGHT INTO THE ROLE OF DISA AND OPEN NEW QUESTIONS IN THE FIELD.

We would like to thank the reviewer for her/his enthusiasm about our biochemical characterization of the interplay of B. subtilis DisA and RecG.

MINOR THINGS:

PAGE6 LINE 246: CORRECT ‘ASSYS’ FOR ‘ASSAYS’

The misspelling was corrected

IN FIGURE1 A), IS A INDICATING THE POSITION OF THE WELLS?; AND IN B), WHAT IS LETTER ‘C’ INDICATING?

In Fig. 1A, the DisA-HJ complexes are retained in the well and denoted as “A” aggregates. In Fig. 1B, “C” is the DNA control without DNase I treatment. The text was amended to answer the query.

PAGE 6 LINE 257 HERE AND LATER THE AUTHORS REFER TO THE KDAPP. THE TEXT WILL BENEFIT IF A BRIEF DESCRIPTION IN THE METHODS, OR AS A SUPPLEMENTARY FIGURE SHOWING THE DATA AND ANALYTICAL METHOD USED FOR ESTIMATING THE CONSTANTS, IS INCLUDED.

In the Material and methods section 2.3 it is now presented how the KDapps were determined

FIGURE 4 LABELS WILL BE MORE LEGIBLE WITH BIGGER FONTS.

As suggested by the reviewer, the font size was enlarged. A new Figure is presented.

PAGE 13 LINE 529 REMOVE EXTRA DOTS FROM: VS…

Done

PAGE 14 LINE 556 REMOVE EXTRA DOTS FROM: VS…

Done

PAGE 16 LINE 645 REFERENCE FORMAT SEEMS WRONG

The unformatted Sinai et al reference was amended

PAGE 16 LINE 663 CORRECT TO ‘WILL BE RELEVANT’

As suggested, the text was modified

Reviewer 2 Report

Review for Cells

Title: DisA limits RecG activities at stalled or reversed replication forks

Transcription replication conflict is a major trait for genomic instability due to the stalling of the Replicative DNA polymerase. Homologous recombination, as well as fork reversal, are conserved mechanisms required for replication restart to bypass DNA damage during replicative stress. Here in this manuscript Torres and colleagues reveal a functional interplay between a DNA damage checkpoint protein deadenylate cyclase DisA and translocase RecG, in the haploid reviving spore of Bacillus subtilis.

RecG is a multifunctional protein and there is conflicting evidence on its model of action. Then, RecG and DisA are in the same epistatic group pointing out their role in the same pathway. Therefore, in this report, the authors propose that: RecG relocates DisA nearby the HJ and inhibits its ATPase activity by trapping it. DisA may downregulate and limits RecG activity in processing of the fork intermediate to prevent fork collapse. Overall, although the paper is difficult to read, the authors gave however a good overview of the problem and the research gap in the field. Despite the strong in vitro evidence, the authors could come up with in vivo data to support and strengthen their model, such as the in vivo colocalization evidence.  To be considered for publication in cells journal,

Minor ccomments:

  • The authors should provide some in vivo data to show that the DisA and RecG can colocalize at the stalling forks? Using immunofluorescence of PLA assay? With or without damage such MMS.

  • The authors may present the coomasie gel of the purified protein DisA and RecG in the main figures.

  • The two complex GI and GII formed by the RecG may be explained by the dimerization of the RecG protein? Or by the structure changing of the fork? (branch migration). This is can be tested by using the ATPase Dead mutant of the RacG, which is a missing control in this experiment. The GII is not observed when you add the DisA? How this can be explained?

  • Figure 1-C, control missing in the figure: RecG plus HJs without DisA.  

  • Is the relocation activity of the DisA by the RecG is dependent on the hydrolase activity of RecG? Does DisA recognize the complex RecG-HJ (before translocation or during translocation (during the moovement)?

  • In the experiment FigureS1-B, the authors should add a control, where they can reconstitute the interaction between DisA RecG and HJ. This can be done by using a biotinylated probe to show the interaction and the formation of the complex. Also, they can assess the formation of the complex in the presence or absence of the ATPase activity of both RecG and DisA (or the dead mutants). This experiment can validate the results of figure 1-A, and show that DisA and RecG, and HJ form a complex in a solution.

Author Response

The referee’s comments are shown in CAPITAL LETTERS  and the answers to their queries in small case letters. The changes are written in a red font in Torres et al wCorrections.

REVIEWER 2

TRANSCRIPTION REPLICATION CONFLICT IS A MAJOR TRAIT FOR GENOMIC INSTABILITY DUE TO THE STALLING OF THE REPLICATIVE DNA POLYMERASE. HOMOLOGOUS RECOMBINATION, AS WELL AS FORK REVERSAL, ARE CONSERVED MECHANISMS REQUIRED FOR REPLICATION RESTART TO BYPASS DNA DAMAGE DURING REPLICATIVE STRESS. HERE IN THIS MANUSCRIPT TORRES AND COLLEAGUES REVEAL A FUNCTIONAL INTERPLAY BETWEEN A DNA DAMAGE CHECKPOINT PROTEIN DEADENYLATE CYCLASE DISA AND TRANSLOCASE RECG, IN THE HAPLOID REVIVING SPORE OF BACILLUS SUBTILIS.

RECG IS A MULTIFUNCTIONAL PROTEIN AND THERE IS CONFLICTING EVIDENCE ON ITS MODEL OF ACTION. THEN, RECG AND DISA ARE IN THE SAME EPISTATIC GROUP POINTING OUT THEIR ROLE IN THE SAME PATHWAY. THEREFORE, IN THIS REPORT, THE AUTHORS PROPOSE THAT: RECG RELOCATES DISA NEARBY THE HJ AND INHIBITS ITS ATPASE ACTIVITY BY TRAPPING IT. DISA MAY DOWNREGULATE AND LIMITS RECG ACTIVITY IN PROCESSING OF THE FORK INTERMEDIATE TO PREVENT FORK COLLAPSE. OVERALL, ALTHOUGH THE PAPER IS DIFFICULT TO READ, THE AUTHORS GAVE HOWEVER A GOOD OVERVIEW OF THE PROBLEM AND THE RESEARCH GAP IN THE FIELD. DESPITE THE STRONG IN VITRO EVIDENCE, THE AUTHORS COULD COME UP WITH IN VIVO DATA TO SUPPORT AND STRENGTHEN THEIR MODEL, SUCH AS THE IN VIVO COLOCALIZATION EVIDENCE.  TO BE CONSIDERED FOR PUBLICATION IN CELLS JOURNAL,

We would like to thank the reviewer for her/his enthusiasm about our biochemical characterization of the interplay of B. subtilis DisA and RecG.

MINOR COOMMENTS:

THE AUTHORS SHOULD PROVIDE SOME IN VIVO DATA TO SHOW THAT THE DISA AND RECG CAN COLOCALIZE AT THE STALLING FORKS? USING IMMUNOFLUORESCENCE OF PLA ASSAY? WITH OR WITHOUT DAMAGE SUCH MMS.

We have previously published in vivo data that suggest this interplay. First, DisA forms dynamic foci in exponentially growing wt cells, but it forms static foci upon recG or recU inactivation (Gándara et al 2017). Here, in the absence of the branch migration translocase RecG or the HJ resolvase RecU, branched recombination intermediates (HJs formed at double strand breaks or reversed forks) accumulate. Second, in sporulating cells, DisA rapidly moves over the nucleoid and upon damage pauses in the wt background or upon inactivation of recJ and addAB, but it maintains its dynamic movement in the absence of recO or recA (Torres et al 2019). The absence of AddAB and RecJ blocks double-strand break repair. Third, essential SsbA is crucial for RecA activation (Carrasco et al 2015), fork remodelling (Torres et al 2019) and replication restart (Seco & Ayora 2017). Finally, the RecG DNA helicases is associate with the chromosomal replication forks by interacting with the SsbA protein (Lecointe et al 2007).  These data suggest that the recombination intermediates recognized by DisA are formed after fork stalling, rather than at collapsed replication forks, and that DisA pausing requires RecO (a RecA loader) and RecA (the recombinase).

THE AUTHORS MAY PRESENT THE COOMASIE GEL OF THE PURIFIED PROTEIN DISA AND RECG IN THE MAIN FIGURES.

In Figure S1B and S1C, the purified DisA (lane 1) and RecG (lane 2) proteins can be seen in a coomasie blue stained SDS polyacrylamide gel.

THE TWO COMPLEX GI AND GII FORMED BY THE RECG MAY BE EXPLAINED BY THE DIMERIZATION OF THE RECG PROTEIN? OR BY THE STRUCTURE CHANGING OF THE FORK? (BRANCH MIGRATION). THIS IS CAN BE TESTED BY USING THE ATPASE DEAD MUTANT OF THE RACG, WHICH IS A MISSING CONTROL IN THIS EXPERIMENT. THE GII IS NOT OBSERVED WHEN YOU ADD THE DISA? HOW THIS CAN BE EXPLAINED?

There are many variables to explain the formation of two protein-HJ DNA complexes with very low resolution. We have no results to support the presence of a hypothetical RecG dimer. Cañas et al (2014) showed that RecG forms very unstable complexes with HJ DNA in the presence of a non-hydrolysable ATP analog, and the complexes are even more unstable in the presence of ATP. Moreover, no RecG-HJ DNA complexes are formed in the absence of ATP. Please remember that we added 0.2% glutaraldehyde prior loading into the gel to stabilize the protein-DNA complexes, and that the protein was incubated with a low hydrolysable ATP variant (experiment analogue to that with an ATPase dead mutant of RecG), and thus no branch migration by RecG was expected. The formation of complex II was also observed and analyzed for the E. coli protein by McGlynn and Lloyd in 2000. These authors demonstrate that this complex II is only observed at high concentrations and that in this case, two independent RecG monomers are bound to the same DNA substrate. We disagree with the reviewer’s comment. When DisA is added, we cannot differentiate if the observed complex is GI or GII, since most DNA remains trapped in the well. We considered that the analysis of these little details will distract the reader from the main goal. Now this information is extended and the correct reference is cited.

FIGURE 1-C, CONTROL MISSING IN THE FIGURE: RECG PLUS HJS WITHOUT DISA.

We have shown that: i) under the conditions used to detect [alpha32P]-c-di-AMP, [alpha32P]-ADP is poorly separated from the [alpha32P]-ATP substrate, therefore, RecG-mediated hydrolysis of ATP cannot be observed (Gándara et al 2017); ii) in the presence of 100 microM ATP, RecG does not significantly hydrolyze ATP (ATP concentration significantly below the Km), while DisA efficiently synthesized c-di-AMP (ATP concentration only 1.5-fold below the Km [Fig. S2]); iii) RecG cannot synthesize c-di-AMP (lane 8); and iv) addition of HJ DNA (lane 3) reduced and addition of both RecG and HJ DNA blocked DisA-mediated c-di-AMP synthesis (lanes 4-6). Since RecG does not show any diadenylate cyclase activity (lane 8) and has no predicted or described domain to do that, we considered that the control in lane 8 is enough, and expect no difference upon HJ addition.

IS THE RELOCATION ACTIVITY OF THE DISA BY THE RECG IS DEPENDENT ON THE HYDROLASE ACTIVITY OF RECG? DOES DISA RECOGNIZE THE COMPLEX RECG-HJ (BEFORE TRANSLOCATION OR DURING TRANSLOCATION (DURING THE MOOVEMENT)?

We are confused, we as well as others have shown that: i) RecG is a motor protein, and it cannot reverse or restore a fork in the absence of ATP hydrolysis; ii) RecG, at preformed RecG-HJ DNA complexes, is only slightly sensitive to DisA regulation of fork reversal or fork restoration, but RecG unwound preformed DisA-HJ DNA complexes with very low efficiency (Fig. 3 C and 3E); iii) DisA inhibits RecG-mediated ATP hydrolysis, but a DisA mutant variant that cannot bind DNA does not inhibit the ATPase of RecG (Fig. 4). As stated in the text and in point (i), the motor cannot translocate in the absence of ATP hydrolysis. Then, we favor that DisA limits RecG before translocation. The text was modified to answer the query.

IN THE EXPERIMENT FIGURES1-B, THE AUTHORS SHOULD ADD A CONTROL, WHERE THEY CAN RECONSTITUTE THE INTERACTION BETWEEN DISA RECG AND HJ. THIS CAN BE DONE BY USING A BIOTINYLATED PROBE TO SHOW THE INTERACTION AND THE FORMATION OF THE COMPLEX. ALSO, THEY CAN ASSESS THE FORMATION OF THE COMPLEX IN THE PRESENCE OR ABSENCE OF THE ATPASE ACTIVITY OF BOTH RECG AND DISA (OR THE DEAD MUTANTS). THIS EXPERIMENT CAN VALIDATE THE RESULTS OF FIGURE 1-A, AND SHOW THAT DISA AND RECG, AND HJ FORM A COMPLEX IN A SOLUTION.

We guess that the reviewer is suggesting pool-down experiments associated to EMSA. We considered to perform this kind of experiments, but they did not allow us to discriminate if both proteins bind to the same or to a different HJ. The experiments presented in Fig. 1B validate the data presented in Fig. 1A. In Fig 1B, using the very sensitive footprinting assay we inferred that DisA, RecG and HJ DNA form a ternary complex.